# Wildfire Dynamics along a North-Central Siberian Latitudinal Transect Assessed Using Landsat Imagery

Yury Dvornikov [1,2], Elena Novenko [2,3], Mikhail Korets [4] and Alexander Olchev [2,5,*]

1   Department of Landscape Design and Sustainable Ecosystems, Agrarian-Technological Institute, Peoples' Friendship University of Russia, 117198 Moscow, Russia; dvornikov_yua@pfur.ru
2   Faculty of Geography, Lomonosov Moscow State University, GSP-1, 1-12 Leninskie Gory, 119991 Moscow, Russia; lenanov@mail.ru
3   Institute of Geography, Russian Academy of Sciences, Staromonetny Lane, 29, 119017 Moscow, Russia
4   V.N. Sukachev Institute of Forest of the Siberian Branch of Russian Academy of Sciences, KSC SB RAS, 660036 Krasnoyarsk, Russia; mik@ksc.krasn.ru
5   Shirshov Institute of Oceanology of the Russian Academy of Sciences, 117997 Moscow, Russia
*   Correspondence: aoltche@gmail.com

**Abstract:** The history of wildfires along a latitudinal transect from forest–tundra to middle taiga in North-Central Siberia was reconstructed for the period from 1985 to 2020 using Landsat imagery. The transect passed through four key regions ($75 \times 75$ km$^2$) with different climate and landscape conditions that allowed us to evaluate regional wildfire dynamics as well as estimate differences in post-fire forest recovery. The Level-2A Landsat data (TM, ETM+, and OLI) were used to derive: (i) burned area (BA) locations, (ii) timing of wildfire occurrence (date, month, or season), (iii) fire severity, and (iv) trends in post-fire vegetation recovery. We used pre-selected and pre-processed scenes suitable for BA mapping taken within four consecutive time intervals covering the entire period of data analysis (1985–2020). Pre- and post-fire dynamics of forest vegetation were described using spectral indices, i.e., NBR and NDVI. We found that during the last three decades, the maximum BA occurred in the southernmost Vanavara region where ≈58% of the area burned. Total BA gradually decreased to the northwest with a minimum in the Igarka region (≈1%). Nearly half of these BAs appeared between summer 2013 and autumn 2020 due to higher frequency of hot and dry weather. The most severe wildfires were detected in the most northeastern Tura region. Analysis of NDVI and NBR dynamics showed that the mean period of post-fire vegetation recovery ranged between 20 and 25 years. The time of vegetation recovery at BAs with repeat wildfires and high severity was significantly longer.

**Keywords:** wildfires; forest tundra; northern taiga; middle taiga; Landsat imagery; Google Earth Engine; North-Central Siberia

## 1. Introduction

Wildfire is a major ecological force influencing long-term vegetation dynamics, ecological functions, and biodiversity [1]. Recent climatic changes, including a rapid increase in global temperature [2], and increase in the frequency of extremely warm and dry weather conditions, have resulted in a higher probability of summer droughts across Siberia [3–8] as well as higher fire risks in the region. Wildfires in Siberia account for 80% of the total number of fires in Russia [9] with the annual burned area (BA) ranging from 2 to 20 Mha [10,11]. Severe wildfires that bring significant ecological and economic damages have become a common feature of fire regimes in coniferous forests worldwide [12,13], particularly in Siberia [14]. Being highly sensitive to climate variability and especially to extreme high temperature events and soil droughts, forest fires also have a significant impact on various atmospheric processes. Forest fires have direct impacts on surface albedo, net radiation, energy, and water fluxes [15]. They also result in large emissions of greenhouse gases (GHG)

and aerosols into the atmosphere [16,17]. All of these factors affect surface air temperature, cloud formation, and precipitation [18].

Since the end of the last century, there were many efforts to map active fires and BAs for all of Siberia using low-resolution satellite remote sensing data such as AVHRR and MODIS [10,19]. The comparison of BA estimates from different data sources, e.g., Krasnoyarsk region (territory corresponding to Central Siberia), have shown up to five-fold differences between these estimates mainly due to various algorithms' specifications [10]. Widely used AVHRR and MODIS data are very useful for regional/global wildfire and carbon sequestration assessments. However, the accurate mapping of BAs and restoration of wildfire history at key sites are often needed for specific research tasks (e.g., analysis of post-fire vegetation recovery [6,20], paleoenvironmental reconstructions, etc.). Long-term satellite data obtained by the Landsat missions allow for detailed analysis of wildfire history since the early 1980s in Central Siberia. The medium spatial resolution (30 m) of Landsat satellite images is well suited for accurate mapping of BAs [21]. Recent development of the Google Earth Engine (GEE) [22] cloud computing platform enables the opportunity to assess the entire Landsat (and other satellites) data archive for fast processing and thematic mapping.

The present study focused on mapping and estimating the spatial and temporal variability of forest fires in North-Central Siberia for the period from 1985 to 2020 using the Level-2A Landsat data. These dynamics were assessed along a latitudinal transect from the forest tundra in the north to the middle taiga in the south. In recent decades, the territory was an area of multi-faceted experimental studies focused on long-term forest dynamics and biodiversity using various ecological and paleo-ecological methods including tree-ring, pollen, plant macrofossil, and charcoal analysis [23,24]. Aggregating modern data on wildfire dynamics with information about fire activity in prior epochs is very important for better understanding the influence of both long- and short-term climate variability and human activity on regional wildfire frequency and severity. The main objectives of our study were to reconstruct the forest fire history, to derive the BA locations and time of wildfire occurrence, and to analyze the fire severity and post-fire vegetation recovery of the key regions within the study area with various landscape and climate conditions. We solved these objectives by answering the following principal questions: (i) Is there any difference in fire activity in different ecotones and forest zones in North-Central Siberia? (ii) Is there any trend in fire activity and fire severity over the period from 1985 to 2020? (iii) What is the main period of post-fire vegetation recovery in the study area?

## 2. Materials and Methods

### 2.1. Description of Key Regions

To describe long-term wildfire dynamics in North-Central Siberia over the last 35 years, four key regions (Igarka, Turukhansk, Vanavara, and Tura) along a latitudinal transect (Figure 1) each with an area of 75 × 75 km were selected. These regions belong to a subarctic climate zone (Dfc) according to the Köppen–Geiger classification with cold summers and uniform precipitation distribution throughout the year [25]. The Igarka key region belongs to forest–tundra ecotone, the Turukhansk and Tura regions—to the northern taiga, and the southern Vanavara region—to the middle taiga forest zone, respectively (Table 1, Figure 1). The vegetation of taiga is mainly represented by larch (*Larix gmelinii* (Rupr.) Kusen., *Larix sibirica* Ledeb.) with admixture of spruce (*Picea obovata* Ledeb.), Siberian pine (*Pinus sibirica* Du Tour), and fir (*Abies sibirica* Ledeb.) in the middle taiga subzone. Some birch species (*Betula pubescens* Ehrh., *B. tortuosa* Ledeb., *B. cajander* Sukaczev) occur in the western and southern regions [26]. The vegetation of the Igarka region is mainly represented by rare woody vegetation with some species of conifers (*Larix sibirica* Ledeb., *Picea obovata* Ledeb. and *Pinus sibirica* Du Tour) that are mixed with shrub thickets (*Duschekia fruticosa* Rupr. and *Betula nana* L.), extensive wetlands, and moss-lichen plant communities. Wildfires are the main driver of forest succession and vegetation change in the key regions [27].

All selected areas are situated in the territory underlain by continuous, discontinuous, or isolated permafrost.

**Figure 1.** Selected key regions along a latitudinal transect in North-Central Siberia (Basemap: ESRI©).

**Table 1.** The location and vegetation zones of selected key regions.

| Key Regions | Region Locations | Vegetation Zone |
|---|---|---|
| Vanavara | 60.3–59.95°N, 101.6–103°E | middle taiga |
| Tura | 64.0–64.7°N, 99.3–101.1°E | northern taiga |
| Turukhansk | 65.5–66.2°N, 87.2–88.9°E | northern taiga |
| Igarka | 67.1–67.8°N, 85.6–87.4°E | forest–tundra |

*2.2. Satellite Data Pre-Processing*

We have divided the ≈35-year timeframe into four consecutive decadal time intervals: 1985–1995, 1995–2005, 2005–2015, and 2015–2020 (Table 2), and for each interval, we found the most suitable Landsat scene from the TM, ETM+, and OLI archives available within GEE [22]. We used the Level-2A processed scenes corrected for atmospheric effects using the LEDAPS algorithm for TM and ETM+ scenes [28] and LaSRC algorithm for OLI scenes [29]. Each selected scene had to meet several requirements for potential BA mapping: cloud coverage less than 5%, taken in July or August, closest acquisition date to the end of the time interval.

**Table 2.** Time intervals for which cloudless Landsat mosaics and SPOT3 (*) scenes were obtained and BA mapped.

| Time Interval | Vanavara | Tura | Turukhansk | Igarka |
|---|---|---|---|---|
| 1985–1995 | 4 July 1994 | 26 July 1995 *<br>21 July 1996 * | 29 August 1987 | 17 August 1986 |
| 1995–2005 | 4 July 2000 | 5 August 2001 | 16 August 1994 | 17 July 1998 |
| 2005–2015 | 24 July 2013 | 29 July 2007<br>20 July 2013 | 18 August 2009 | 23 August 2009 |
| 2015–2020 | 2 July 2020 | 30 July 2020 | 21 August 2019 | 18 July 2019 |

July and August were chosen, since the maximum wildfire frequency in the north latitudes 60–70°N in Central Siberia is usually observed during these months [30]. Moreover, the period is completely free of late/early snow cover that sometimes can be present on scenes acquired in June and September.

A single scene was often unable to cover the entire key region area. Therefore, we have mosaicked acquisitions taken within the five neighboring days.

For the Tura region, there were no suitable Landsat TM and ETM+ scenes taken earlier than 2000. Therefore, for this region, we additionally used the local forest inventory data for 1989, SPOT-3 acquisitions taken during July 1995/1996 (the data acquired by CNES's Spot World Heritage Programme) (Table 2) and Global Forest Change 2000–2019 data [31].

### 2.3. Burned Area Mapping

We manually delineated all BAs from the false-color composites of selected scenes and stored them as vector data. These vector BAs were further used to define the date of wildfire occurrence (or year, depending on the data availability) by detecting abrupt changes in the NBR time series within GEE. For large wildfires that occurred after 2000, we also used the 500 m MODIS Burned Area product (MCD64A1 version 6, NASA LP DAAC) for date establishment. We further defined areas that were burned twice or three times by overlaying BAs of different time intervals.

### 2.4. Fire Severity and Vegetation Recovery

We retrieved interannual trends (median values) of several spectral indices including the NBR [32,33], NDVI [34], and Modified Normalized Difference Moisture Index (MNDMI) [35], for each detected BA. MNDMI was used for filtering the NBR and NDVI data retrieved under waterlogged conditions: i.e., we excluded outliers of index (NBR, NDVI) values if MNDMI exceeded two standard deviations calculated for each MNDMI trend to avoid values retrieved from images taken under, for instance, extremely wet conditions (after heavy rain, etc.). Cleaned NBR and NDVI trends were used to (1) assess the fire severity; (2) describe vegetation dynamics at each BA; (3) estimate the succession rate for each BA. Fire severity was assessed by differenced NBR (dNBR) index values [36] as:

$$dNBR = \text{pre-fire NBR} - \text{post-fire NBR.} \qquad (1)$$

We analyzed succession rate and vegetation dynamics by comparing NDVI trends of burned and unburned forest sites.

### 2.5. Statistical Processing

For the wildfire history analysis, we summarized the number of wildfires as well as the total area burned for each year. For the vegetation recovery assessment, we summarized the NDVI values of BAs by 5-year intervals starting from 1985 and compared them with NDVI values at undisturbed sites. We used paired T-test for finding statistically significant differences between NDVI of pixels belonging to BA and undisturbed sites for each time interval. The significance of trends was assessed by the Mann–Kendall non-parametric test [37]. The statistical processing was performed in R environment [38].

## 3. Results

### 3.1. Wildfire History along the North-Central Siberian Latitudinal Transect

Analysis of Landsat data for four selected key regions (with a total area of 22,500 km$^2$) allowed us to detect a large area (up to 4612 km$^2$ or 21% of total area) damaged by wildfires from 1985 to 2020, which was unevenly distributed over time. The maximum wildfire activity was observed in 1990, 2006, 2013, and 2016 with the number of fires totaling 17, 5, 10, and 12, respectively. Wildfire BA summed across the three most extreme fire years (1990, 2006, and 2016), totaled 2666 km$^2$ or about 58% of the total BA over the entire study period (Figure 2). There was no statistically significant trend in the total area burned in the

period from 1985 to 2020 (Mann–Kendall's tau = −0.07, p = 0.617) nor for the total number of wildfires (Mann–Kendall's tau = 0.04, p = 0.813).

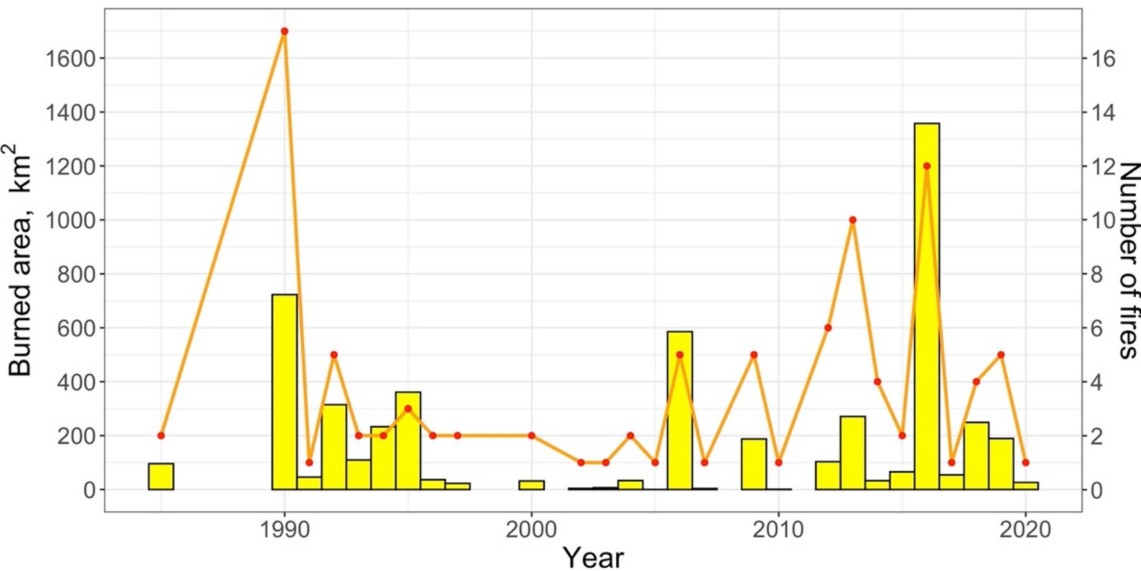

**Figure 2.** Temporal variability in total BA and number of wildfires across the four key regions. Total burned area—bar chart. Number of wildfires—line chart.

### 3.1.1. Vanavara Region

For the Vanavara region, we were able to retrieve the exact fire dates after the spring–summer season of 1991. Seventeen BAs with a total area of 874 km$^2$ were detected within the Vanavara region on the Landsat TM image on July 4, 1994 (Figure 3a). Most of these BAs (n = 12) occurred before September 30, 1991 and had a total area of 602 km$^2$ (69%). Considering the weak post-fire vegetation regeneration at these BAs, they likely occurred no earlier than in the summer season of 1990. Four BAs with the total area of 269 km$^2$ (31%) occurred within the time period from September 30, 1991 to August 31, 1992, which was likely in the summer period of 1992, since there is no evidence that wildfires had occurred in late September in this area. A small BA with an area of ≈3.1 km$^2$ occurred in the summer of 1993.

An analysis of the Landsat TM image taken on 4 July 2000 showed that the total BA did not significantly increase in the region after 1994 (Figure 3a). Three small BAs with a total area of 29 km$^2$ were detected along the left and right banks of the Podkamennaya Tungusska River; one of them occurred in July of 1994 and two others occurred during the summer of 1997. In the following years, a severe wildfire occurred on the left side of the river in the same area where the forest fire of 1990–1991 had been detected. The total area damaged by this fire was 227 km$^2$, including 114 km$^2$ that were previously burned. There was no evidence of the wildfire severity in the recently burned area. By 4 July 2000, the total BA reached 986 km$^2$ (17.5% of the Vanavara region) and increased by 112 km$^2$ compared to 1994.

In the period between July 2000 and July 2013, 451 km$^2$ of the study area burned, including 372 km$^2$ of new areas that were previously not affected by wildfires (Figure 3a). Repeated forest fires within the recently burned sites (1991–1992 and 1994–1995) occurred across ca. 79 km$^2$; the oldest BAs occurred in August 2005 and the most recent occurred in July 2012. Seven small BAs with a total area of 74 km$^2$ occurred between July–August 2012 and June–July 2013. Two very small BAs (10 km$^2$) appeared in 2002 and 2007, and two more large BAs with a total area of 300 km$^2$ occurred in the summer (July–August) of 2006 in the western and eastern parts of the study area. Despite the very large forest areas damaged by these fires, they did not affect the recently burned area (1991) and mainly impacted older

forests. Moderate wildfire activity occurred in July–August 2012 close to Vanavara village (less than 7 km) but was stopped by a fire trench in the northeast direction.

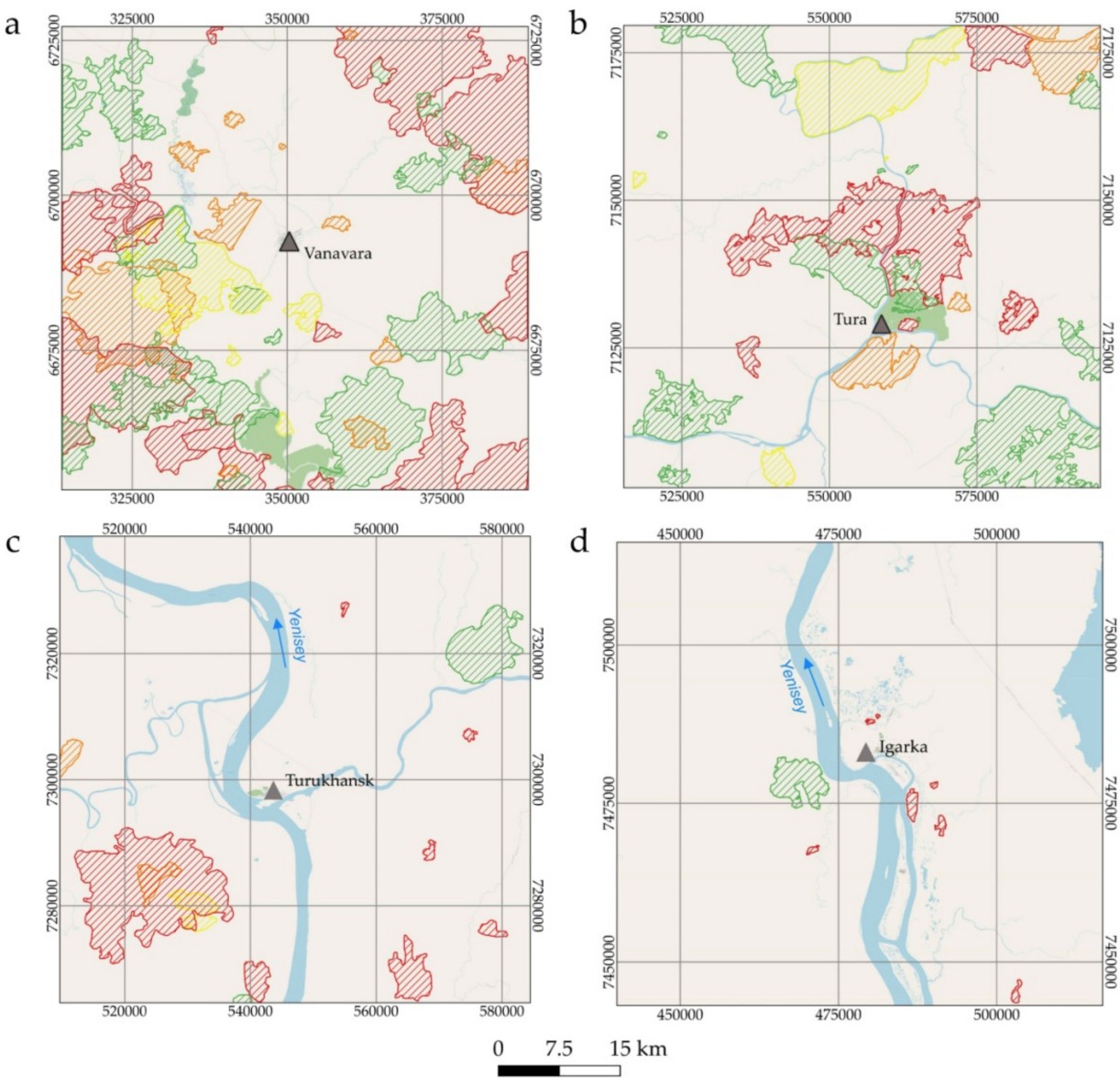

**Figure 3.** Burned areas within the key regions detected from Landsat scenes: (**a**) Vanavara, green: 4 July 1994; yellow: 4 July 2000; orange: 24 July 2013; red: July 2020; (**b**) Tura, green: 5 August 2001; yellow: 29 July 2007; orange: 20 July 2013; red: 30 July 2020; (**c**) Turukhansk, green: 29 August 1987; yellow: 16 August 1994; orange: 18 August 2009; red: 21 August 2019; (**d**) Igarka, green: 17 July 1998; red: 18 July 2019. Basemap: OpenStreetMap©. Datum: WGS-84, Projection: Universal Transverse Mercator Zone 48N (**a**), Zone 47N (**b**), Zone 45N (**c**,**d**). Black triangles show locations of the district centers.

In the period from 2013 to 2020, the total BAs in the Vanavara region significantly increased; seventeen new BAs were found, with the total BA of 1356 km$^2$ (Figure 3a). Although this period was half as short (7 years) as the previous period (13 years), the area damaged by wildfires was twice as large. Most of these new BAs (1200 km$^2$) were detected in new territories, resulting in a doubling of the total BA within the study region compared to 2013 (from 1340 km$^2$ to more than 2500 km$^2$ by July 2020 or 45% of the total region area). In 2013–2020, wildfires affected the southwestern part of the region with

maximum forest damage on the left bank of Podkamennaya Tungusska River, as well as in the southeastern and northeastern parts of the study region. Most of these new wildfires occurred in July–August 2016 (six BAs with a total area of 983 km$^2$). Another area of 65 km$^2$ burned in July–August of 2013 and 2014. One wildfire (54 km$^2$) occurred in August 2017, two occurred in July–August 2018 with the total burned area of 177 km$^2$, and three occurred in July–September 2019 (76 km$^2$). New wildfires have also partly affected the previously burned (1991–1992 and 2006) areas.

### 3.1.2. Tura Region

The analysis of BAs from 1990 to 2000 (Figure 3b) was based on local forest inventory data (1989) and SPOT3 images (July 1995, 1996) and showed a clear decreasing trend in the area and number of BAs in the region: 1990–1998 (number of wildfires (n) = 11, BA = 643 km$^2$); 2000–2007 (n = 6, BA = 302 km$^2$); 2009–2014 (n = 5, BA = 370 km$^2$); and 2015–2020 (n = 7, BA = 297 km$^2$). The total BA during the period from 1990 to 2020 in the region was 1579 km$^2$ (24.5% of the total region area), excluding the overlapping BAs (repeated fires) of 34.7 km$^2$.

### 3.1.3. Turukhansk Region

Most of the area around Turukhansk (412.4 km$^2$) burned between 2004 and 2020. Only three relatively old BAs were found: two wildfires that occurred earlier than the summer 1985 with a total area of 95.8 km$^2$ and one fire in the summer of 1990 that burned 27.4 km$^2$. These fire events were followed by a ≈15-year fire-free period. Two forest fires occurred in the summer of 2004 (33 km$^2$). Since then, wildfires only occurred during 2013–2016 (seven wildfires with the total area of 379 km$^2$). The largest wildfire (308 km$^2$) in the area occurred on the left bank of the Yenisei River in July 2016 and affected areas that previously burned in 1990 and 2004 (Figure 3c). The total BA in the study region was 535.6 km$^2$ (9.5% of the total region area).

### 3.1.4. Igarka Region

Only eight BAs were found within the Igarka region for the entire study period from 1985 to 2020 (Figure 3d). The largest forest fire (46 km$^2$) occurred on the left bank of the Yenisei River between July 28, 1988 and August 17, 1992 (there were no data to determine exactly the time of wildfire occurrence). Seven other small BAs with a total area of 15 km$^2$ occurred in August 2009, summer of 2010, July 2013, and summers of 2014 and 2016. The total BA did not exceed 1.1% of the total area of study region (Figure 3d). Thus, this territory was the least exposed to forest fires.

### 3.2. Temporal Dynamics of NDVI, NBR and Fire Severity

Analysis of post-fire vegetation recovery in most BAs showed positive post-fire NDVI dynamics for all study regions. We found that the threshold value of NDVI (0.5–0.6) for 25 forest BAs occurred between 1990 and 1995 across all study regions and reached the mean background NDVI values of undisturbed forest sites (0.75–0.85) in 2015–2020 ($p < 0.01$) (Figure 4). Therefore, the period of 25 years can be considered as a period of partial vegetation recovery in all study regions. The background pre-fire NBR values of undisturbed forest (median = 0.64) was found to drop to 0.4 after fire at some representative BA (e.g., wildfire of 1992). In 2015, the mean NBR value for this BA again reached 0.6. The same trend was observed for NDVI; in 2015, the mean NDVI of the BA returned to its initial value (0.8).

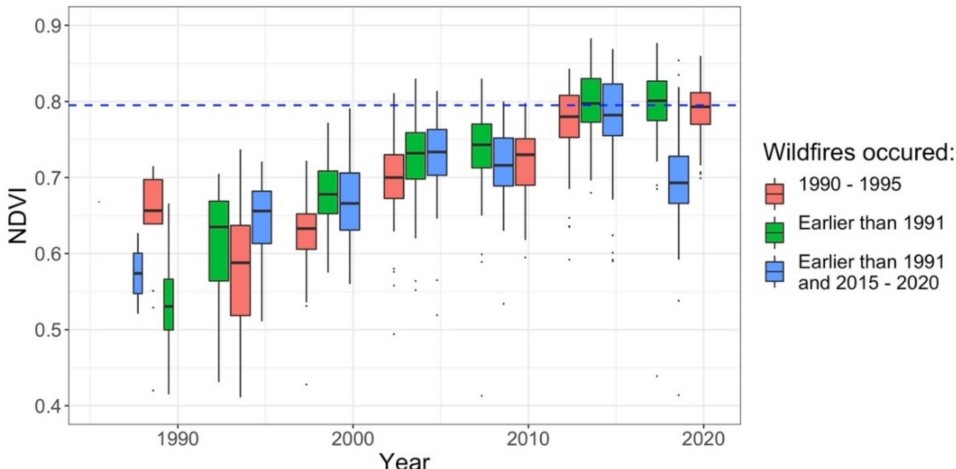

**Figure 4.** Temporal NDVI variability of Bas burned in the periods (i) between 1990 and 1995; (ii) earlier than 1990; and (iii) twice burned: before 1990 and during the period between 2015 and 2020. Dashed blue line represents median NDVI across undisturbed areas within four key regions (number of undisturbed sites: 31).

The rate of post-fire forest recovery slows down significantly at BAs where wildfires returned. Within the Vanavara region, we found several BAs where fires occurred three times during the study period. The mean NBR values for two of these BAs were 0.38 and 0.33 after the first fire in 1990 or earlier. Fifteen years later (summer 2006), one of these BAs again burned, and NBR and NDVI values dropped down from 0.47 and 0.67 to 0.21 and 0.59, respectively. The NBR and NDVI values of forest sites burned in 2012 did not return to initial values of ≈0.6 and ≈0.8 in 2020 and were 0.50 and 0.71, respectively. The NBR and NDVI values for the forest sites that burned several times in 2012 and 2016 decreased from initial (NBR: 0.59, NDVI: 0.73) values to 0.43 and 0.66 in 2020, respectively.

Analysis of dNBR variability at all BAs shows that all study regions are mainly characterized by low ($0.1 \leq$ dNBR $< 0.27$) and moderate ($0.27 \leq$ dNBR $< 0.66$) fire severity with a median value of 0.3 (Figure 5). In the Igarka, Turukhansk, and Vanavara regions, the median fire severity was close to the total median value (0.32 across 7 fires, 0.35 across 11 fires, and 0.28 across 52 fires). The Tura region is characterized by the most severe fires (dNBR: 0.54 for 21 total fires). Five fires had a very high severity (dNBR $\geq 0.68$), whereas only one wildfire of high severity (dNBR = 0.68) was detected in the Vanavara region (July 2017) (Figure 5).

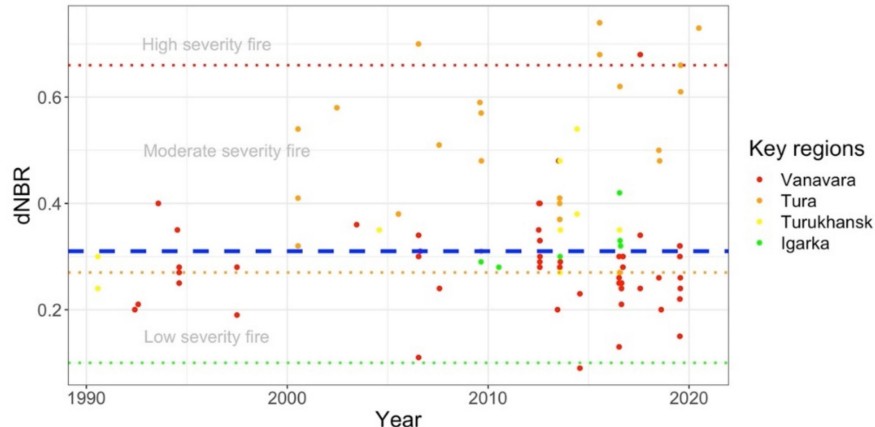

**Figure 5.** Differences of NBR (dNBR) observed within BAs (n = 91) across all study regions. For recurrent wildfires, we consider fire episodes as having a fire return interval of less than 20 years. Blue dashed line is the median dNBR value for all BAs.

## 4. Discussion

### 4.1. History of Wildfires in North-Central Siberia Regions

Forest losses caused by wildfires in Russia account for up to 65% of all forest loss caused by natural and anthropogenic factors [39]. Recent studies have shown that the total number of wildfires and BAs in Siberia has increased by about six to eight times from 1996 to 2015 [30]. This increase is mainly influenced by the increased frequency of extreme high-temperature events with prolonged periods without precipitation that often result in summer droughts. Our analysis of number of fires and area burned (Figure 2) for the selected key regions in North-Central Siberia did not reveal a statistically significant trend neither for total burned area nor for the total number of wildfires. There were several anomalous years with extremely hot and dry weather in summer (e.g., 2016) that caused a strong increase in the number and severity of forest fires in the regions. The total number of wildfires within the four selected key regions account for only 0.45% of all Central Siberian fires that occurred in the larch dominated landscapes between 1996 and 2015 (n=22400) [40]. This is expected when considering the maximum wildfire activity recorded in the more southern latitudes (52–53°N). The low number of detected wildfires within the most northern Igarka key region can be also explained by low temperatures, low frequency of dry thunderstorms, poor availability of woody biomass and litter as fire fuel, as well as high waterlogging of the area. According to recent estimates of the spatial distribution of forest fires in the Central Siberia, less than 1% of all wildfires occurred at latitudes similar to the Igarka region [40]. The high frequency of forest fires in the Vanavara region can be explained by the high frequency of thunderstorms in the summer seasons and the small area of unburnable wet habitats compared to Igarka or Turukhansk regions [27]. The analysis of BA distribution in the forest–tundra zone of Western Siberia also revealed 1989/1990 and 2017 (for Central Siberia: 2016) as years with the highest forest fire activity [41]. The year of 2006 was extreme for the entire Krasnoyarsk region when the areas of forest fires varied from $0.5 \times 10^6$ to $3.0 \times 10^6$ ha [10].

Considering the different environmental factors influencing fire activity in Central Siberia, the spatial distribution of permafrost must be also considered. The study areas are underlain by continuous, discontinuous, or isolated permafrost that can influence the spatial vegetation distribution and surface moisture conditions either directly or indirectly, and they can in turn impact forest fire spatial patterns and severity [18]. At the same time, forest fires can also be an important driver of permafrost thaw that can intensify the emission of GHG into the atmosphere and accelerate global warming [7].

Information about wildfires in the past decades is not limited to remote sensing data only. There are many other sources of information, including historical chronicles, forest inventories, dendrochronological, and paleo-ecological data [7,40,42] that can be used as an alternative to remote sensing data to reconstruct the fire history in the study region for a time interval much longer than the period for that the satellite data are available. Burnt timber is often found in the soil cuts at the depth of the permafrost layer. There are also some historical chronicles reporting forest fires in the region [7,43]. This information can help to roughly estimate the timings of wildfire occurrence without exact assessment of the areas or the number and severity of forest fires.

Available past forest inventory maps (1953–1955) for the Tura region registered large fires in 1947 and 1951 that covered an area of more than 370 km$^2$ (10% of the study area). In particular, the areas damaged by forest fires in the period from 1960 to 1970 exceeded 3500 km$^2$, suggesting more area burned during this period than during any other period of the same duration from 1985 to 2020. Accurate reconstruction of the fire history using various methods can help to obtain information about the long-term variability of fire activity in the study area to estimate the existing relationship between fire frequency and severity, local human activity, and regional climate conditions. Such information can be useful for better understanding the processes and factors that are responsible for spatial wildfire distribution in the study area and can also help for future projections of fire activity in the Central Siberia. Modern climate changes accompanied by

warming, changes in precipitation, and the increasing trend of extreme heat waves and droughts [8] can significantly increase the risk of wildfire occurrence. Thus, information on fire activity and climate changes in the past epochs can be important for developing reliable future projections.

*4.2. Post-Fire Vegetation Recovery and Fire Severity*

The rate of post-fire vegetation recovery is influenced by various factors including the pre-burn vegetation, fire types (ground, crown, or surface fires) and severity [44], and recovery can take several decades to centuries [20]. Some studies report the influence on vegetation recovery by local site conditions including surface topography (slope and aspect), ground water level, and permafrost [45]. Surface topography affects the local microclimatic (e.g., solar radiation, air and soil temperature) and soil moisture conditions influencing the primary production and transpiration rates of recovered vegetation. Available field data for experimental areas, as well as results from our dNBR analysis (Figure 5), indicate fires tend to be low to moderate severity and include a large proportion of surface fire. Such fires that occur in areas with a thin litter layer and a sparse understory can often result in insignificant damage to the woody vegetation, making these fires difficult to recognize from satellite images.

Despite the large BAs in the Vanavara region, the most severe wildfires were detected in the northeastern part of Tura regions mainly due to the higher climate continentality associated with higher precipitation deficiency and the large amount of high fire hazard larch trees in the forest canopy. The various fire severity in the regions, along with different climate conditions, can lead to different periods of post-fire vegetation recovery even within the same study region. Considering the observed significant difference of the recovery periods among the study regions, we focused on the general post-fire recovery pattern and analysis of specific fire events. The results showed that the median NDVI values of BAs within all study regions returned to the levels of unburned sites (0.75–0.85) in a period of about 20–25 years (Figure 4). Considering the moderate severity of wildfires at our study regions, such time of partial vegetation recovery can be considered as plausible. Existing data on the larch forest recovery in the Southern Siberia show that the highest rates of forest recovery were observed within moderately burned patches of a single BA [6]. Analysis of NDVI values for BAs in the tundra of Western Siberia showed that the peak NDVI values in the sites that burned 28 years ago even exceeded the NDVI of non-disturbed sites [46]. Similar results were reported in the study of the post-fire recovery of forest–tundra ecotone in Canada showing that the spectral recovery of burned vegetation was observed after 22 years [47], agreeing with estimations made in our study.

In a study conducted by Knorre et al. [43], it was shown that the vegetation recovery in the polar climate of Siberia is strongly correlated with the depth of the permafrost active layer. The thawing of the upper permafrost stratum caused by wildfires results in a strong increase in the active layer thickness and a decrease in the upper permafrost horizon. The recovery of the permafrost active layer starts after the first ≈20 years following fire due to active vegetation recovery that reduces incoming solar radiation to the soil surface and the heat transfer into the deep soil horizons.

In our study, we focus on wildfire analysis of several key regions along a North-Central Siberian transect from Igarka in the arctic forest–tundra ecotone to the Vanavara region situated in the middle taiga zone. Various landscape properties and climatic conditions can explain some differences discovered in the spatial and temporal patterns of wildfires. The question of the possible reasons for wildfires was beyond the scope of this study. Neighboring settlements and villages pose a relatively high risk of human-induced fires [48]. It can be expected that the areas situated far from settlements can have other features of spatial and temporal BA distributions due to both low risks of human-induced fires and the lack of any opportunities to quickly fight and extinguish the forest fires, resulting in more significant forest losses due to fires in remote regions.

## 5. Conclusions

Wildfire history was reconstructed, and post-fire vegetation recovery was estimated from 1985 to 2020 using Landsat data for four key regions in North-Central Siberia of Russia from the forest–tundra ecotone to the middle taiga zone (total area of 22,500 km$^2$). Our analysis allowed us to detect large areas (up to 4612 km$^2$ or 21% of total area) that were impacted by wildfires from 1985 to 2020. Maximum wildfire activity was observed in 1990, 2006, 2013, and 2016 with the number of fires 17, 5, 10, and 12, respectively, and were mainly associated with very warm and dry weather conditions in the region. The total burned area of the three most extreme years (1990, 2006, and 2016) amounted to 2666 km$^2$ or approximately 58% of the total burned area over the entire study period.

The selected regions are characterized by various landscape properties and climate conditions that allowed us to explain some of the differences discovered in spatial and temporal wildfire distributions. The smallest areas of wildfires were detected in the two most northwestern regions: Igarka and Turukhansk. The highest frequency of forest fires was detected in the most southern Vanavara region. Despite the large areas burned in the Vanavara region, the severity of all wildfires was close to the fire severity detected in the northern Igarka region. The maximum severity of wildfires was found in most northeastern Tura region.

Estimations of vegetation recovery rates for all selected area showed that median NDVI values of burned areas returned to the level of unburned sites (0.75–0.85) in a period of roughly 20–25 years. In many cases, vegetation recovery indicates the partial recovery of woody vegetation in burnt areas only. The total forest recovery usually requires more time depending on the degree of forest damage by fire and can be influenced by multiple landscape and climate factors. The results of this study are in agreement with available literature and could be very useful for future analysis and projections of wildfire frequency and severity in Central Siberian regions under future climate change.

**Author Contributions:** Y.D. and M.K. performed experiments and data analysis, Y.D., E.N., M.K. and A.O. wrote the paper. A.O. edited and finalized the manuscript. All authors have read and agreed to the published version of the manuscript.

**Funding:** The studies of the key study regions in Vanavara, Igarka, and Turukhansk (field experiments and satellite data analysis conducted by Y.D. and E.N.) were supported by the Russian Science Foundation (grant 20-17-00043). The data analysis for the study area in Tura was conducted by M.K. and supported by the Russian Science Foundation (grant 21-17-00163). The forest inventory data analysis was also conducted by M.K. and supported by the Russian Foundation for Basic Research (grant 20-45-242908).

**Data Availability Statement:** Not applicable.

**Acknowledgments:** We thank EcoSpatial Services L.L.C. (USA) for fruitful comments and English edits on the manuscript. The satellite data analysis and statistical processing provided by Y.D. were promoted by the Strategic Academic Leadership Program of RUDN University (Russia).

**Conflicts of Interest:** The authors declare no conflict of interest.

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
