# Peer review of "Wildfire Dynamics along a North-Central Siberian Latitudinal Transect Assessed Using Landsat Imagery"

_remotesensing, doi:10.3390/rs14030790_

Round 1

Reviewer 1 Report

Authors addressess an interesting topic. However, my major concerns are (I) the use of a low number of regions and years when using GEE, (II) the lack of rigor and scientific soundness, even not having statistical analysis and (III) the low quality of figures. Thus, I recommend to re-make or to largely improve the manuscript.

Some line-by-line recommendations are below:

L21-22: according to the absence of clouds? Presence of burned area? What is the GEE algorithm for?

L44: Please, be rigorous when citing articles. For instance, Natole et al. neither talk about fire severity, nor investigate fire severity in Siberia. Check the appropriateness of all references in the manuscript.

L45: This paragraph is superfluous as it is not clearly linked with your work. Please, be concise.

L65: Explain and structure better your objectives. If your objective is to analyze temporal and spatial patterns of burned areas, why you study fire severity and regeneration too?

L70: In Central Siberia.

112: manually mapped using the 112 GEE platform sounds contradictory. Have you manually painted polygons over the burned areas in GEE? Have you painted it over false color composites? Or on the contrary you used some algorithm? Please, clarify the BA method.

L131: What about the rest of years? Are they discarded because the absence of fires?

135: I recommend to separate BA mapping (2.2.) and burn severity mapping (2.3) in two different subsections, providing more details of both. (i.e.: specify algorithms used for spectral indices calculation).

L136: I miss a statistical analysis of the data

L157: What is the meaning of the black square within the figure? Is the study área? If so, whi some burned areas are out of it?

L239-231: These lines are about results shown in the previous section.

L234: What is the burn severity trend? Have you performed a statistical analisis? E.g. Mann kendall, theil shen or linear model?

L235: The quality of this figure, and in general of all figures is quite low for an internationa SCI journal. Revise the vertical axis title.

Reviewer 2 Report

The authors take use of Landsat archive to retrieve historical wildfire and examine the burnt area (BA) in four regions. I think it is useful but the presentation is not clear. I am particularly confused about the method for deriving BA. The authors mentioned "BA was manually mapped". But how BA was mapped. Only several discrete years were selected and how can those years be representative for each period? Another concern is that there is very little work related to remote sensing technique. 
More detail as follows:
Line 115 Page 3: It is unclear how NDVI, NBR and MNDWI were derived from Landsat or/and SPOT3 images. Furthermore, those abbreviations should be fully named when they appear for the first time.
Line 133 Page 3: It is not clear how BAs were manually mapped from "For all selected years BAs were manually mapped using the GEE platform"
What are the scenes used for in Table 2? The title of the table mentioned "BA were mapped" but how BA were mapped? Also, the title of Table 2 "Time periods for which cloudless Landsat and SPOT3 (*) scenes", the scenes number, row/col should be provided. How many scenes are selected for each region for each period? If a single scene is selected for a region, does it cover the whole area? For Tura, it seems one scene cannot cover the complete area. Is it possible that some of the four  selected regions are located between scenes and thus need several scenes to masaic?
Line 121 Page 4: "The BA vector data were stored in the PostgreSQL/PostGIS©..." where are those vector data come from? Are they from the manually mapped BA? 
Line 137 Page 4: In section 3.1, the wildfire occurence for the 4 regions are described in detail. It looks like all description comes from statistics rather than from analysis of remote sensed imageries, i.e., Landsat series. I feel those description, each by each, can be improved because the current version provides no useful information. A table summerizing the result for the 4 regions is enough and more focuses should be on how wildfire were extracted from remote sensing images.
Line 228 Page 8: section  3.2 maps the three indices and it is not clear what temporal resolution is applied. Are they anual max, mean, median?
Figure 6 should be in section 3.1 as this figure shows the number of wildfires and statistical result of BA.
MNDWI was not presented in this section but the method (and abstract) mentioned "...obtain information about temporal variability of NDVI, NBR and MNDWI for each BA" (Line 115 page 3).
English needs improvement, for example, 
Line 133 Page 4: "There is only one suitable scene for the time period 1995-2005 for Turukhansk region and was obtained on 1994-08-16"

Round 2

Reviewer 1 Report

The authors have largely improved the manuscipt, and particularly the methods section. However, I suggest some important but rapid changes before acceptance:

L67 to 83: In this paragraph the objective/s are not clearly stated. The main goals should be stated according to my comment in the previous round other (you study severity and regeneration, but these are not your objectives in the current form).  Instead, the authors use this paragraph to describe study sites and methods (i.e. what they did and how), that should be placed in the next section.

L94: authors of plant species should be included the first time they are mentioned in the document. Example: Larix decidua Mill.

Once again I suggest to improve figures 4, 5 and 6: for instance, Fig. 5 has a different background color than Fig. 4 and 3. being the same type of figure. Scale bars show different styles. The frame of the map should be adjusted to the black frame of study region to save unnecesary space. Also, I recommend to join the 3 figures (3,4 and 5) in a one horizontal panel. In the current form they give a "technical report aspect" to the document rather than a scientific article.

Moreover, Fig. 7 and 8 should have the same horizontal axis title: Time (year) or just Year. But please, be consistent. Same legend position would be nice too for both figures.

Author Response

We are very thankful to reviewer for very helpful and constructive comments and recommendations. The manuscript has been revised in accordance with made suggestions. Below we present the point-by-point answers to the reviewer’s comments.

L67 to 83: In this paragraph the objective/s are not clearly stated. The main goals should be stated according to my comment in the previous round other (you study severity and regeneration, but these are not your objectives in the current form).  Instead, the authors use this paragraph to describe study sites and methods (i.e. what they did and how), that should be placed in the next section.

The last paragraph of introduction was completely revised. Provided information about study sites and methods used was removed or shifted to the method part. We add more complete information about the main objectives of the study, as well as the key scientific questions that is solved in our study.

" The main objectives of our study were: to reconstruct the forest fire history, to derive the BA locations and time of wildfire occurrence, to analyze the fire severity and post-fire vegetation recovery of the key regions within the study area with various landscape and climate conditions. We solved these objectives by answering the following principal questions: i) is there any difference in fire activity in different ecotones and forest zones in North-Central Siberia? ii) is there any trend in fire activity and fire severity over the period from 1985 to 2020? iii) what is the main period of post-fire vegetation recovery in the study area?"

L94: authors of plant species should be included the first time they are mentioned in the document. Example: Larix decidua Mill.

Full Latin names of plant species were added to the article (in Method part at places where they were first mentioned).

Once again I suggest to improve figures 4, 5 and 6: for instance, Fig. 5 has a different background color than Fig. 4 and 3. being the same type of figure. Scale bars show different styles. The frame of the map should be adjusted to the black frame of study region to save unnecessary space. Also, I recommend to join the 3 figures (3,4 and 5) in a one horizontal panel. In the current form they give a "technical report aspect" to the document rather than a scientific article.

All maps were prepared with much higher spatial resolution. We used the same color background (at least the difference is now not visible in both word and pdf documents). The frames of maps were adjusted to black frames.  All maps were aggregated within one figure.

Moreover, Fig. 7 and 8 should have the same horizontal axis title: Time (year) or just Year. But please, be consistent. Same legend position would be nice too for both figures.

The title of horizontal axis and legend positions were unified for both figures.

Reviewer 2 Report

I think the authors have addressed my concerns and now I agree publication.

Author Response

We are very thankful to reviewer for very helpful and constructive comments and recommendations.